

# An evaluation of migration fidelity of Ruby-throated Hummingbirds inferred from stable isotope methods

Geoff Koehler[1], Kevin J. Kardynal[2], Ron E. Jensen[3] and Keith A. Hobson[1,4,†]

[1] NHRC Stable Isotope Laboratory, Environment and Climate Change Canada, Saskatoon, SK, Canada
[2] Science and Technolgy Branch, Environment and Climate Change Canada, Saskatoon, SK, Canada
[3] King Cres, Saskatoon, SK, Canada
[4] Department of Biology, University of Western Ontario, London, ON, Canada
[†] Deceased.

Corresponding author
Geoff Koehler, geoff.koehler@usask.ca

## ABSTRACT

Knowledge of spatial connectivity between breeding and non-breeding locations of migratory birds and their breeding site fidelity are important for avian conservation. Ruby-throated Hummingbirds (RTHU, *Archilochus colubris*) breed in eastern Canada west to the Rocky Mountains and in the USA east of the Mississippi River and spend the non-breeding period in Mexico, Central America, and southern Florida, USA. We measured the hydrogen and oxygen stable isotopic compositions of adult RTHU tailfeathers (fourth rectrix) from three breeding locations in North America to estimate migratory connectivity between breeding and non-breeding grounds where feathers are grown. Feather $\delta^2$H values showed no statistical difference among the three sampling locations as well as disparate geographic assignments from one location on the non-breeding grounds in Costa Rica. Therefore, only weak evidence of migratory connectivity between breeding and non-breeding grounds could be ascertained for our sample of this species. The lack of migratory connectivity detected for Ruby-throated Hummingbirds using stable isotopes is consistent with origins from broad regions on the non-breeding grounds. However, it may also imply that precipitation $\delta^2$H values on the non-breeding grounds do not vary enough to detect a difference among our study populations. Sampling of additional populations in the eastern and southern portion of the species' breeding range and the non-breeding grounds may reveal differences in migratory connectivity among populations and requires further investigation. The $\delta^{18}$O values of feathers correlated poorly to their $\delta^2$H values, an effect that may reflect the balance between metabolically driven processes and environmental water on the $\delta^{18}$O values of hummingbird tissues. This study provides the foundations for further investigations into migratory connectivity of RTHU using $\delta^2$H$_f$ values and suggests potential avenues of study for use of $\delta^{18}$O values of tissues in metabolic research.

## INTRODUCTION

North American bird populations have declined significantly in recent decades, primarily as a result of human activities (*Rosenberg et al., 2019*) and, like other birds, populations

of hummingbirds have similarly declined (*English et al., 2021*). This is important because hummingbirds play a critial role as pollinators and are essential to ecosystem function in many environments (*Leimberger et al., 2022*). Ruby-throated Hummingbirds (RTHU; *Archilochus colubris*) are the most common and widely distributed North American hummingbird and travel thousands of kilometres annually between their breeding grounds in east-central Canada and the USA and their non-breeding grounds in Mexico, Central America, and southern Florida USA (Fig. 1). It has been estimated that breeding populations of this species have declined by about 17% throughout their range since 2004 (*English et al., 2021*). While these declines have been observed on the North American breeding grounds, they may also reflect detrimental changes on the wintering grounds, such as habitat loss and climate change (*Toledo-Aceves et al., 2011*).

Research into the spatial linkages between breeding and non-breeding grounds, or migratory connectivity, is an important parameter for the effective conservation of migratory birds (*Webster et al., 2002*; *Boulet & Norris, 2006*; *Procházka et al., 2008*). Conventional marking methods (*i.e.* banding, ringing) used to study migratory connectivity of birds are historically ineffective for small birds, including hummingbirds, because recapture rates are low and they are generally too small to be fit with tracking devices (*Zenzal Jr, Diehl & Moore, 2014*).

Stable isotope methods, however, have been particularly useful estimating migratory connectivity for birds. This is because the hydrogen and oxygen stable isotopic compositions of feathers are directly related to those of local precipitation, which follow a relatively well-defined geographic gradient (*Craig, 1961*; *Rozanski, Araguás-Araguás & Gonfiantini, 1993*). This allows estimates of the geographic origins of migratory species when captured away from the region of growth (*Hobson & Wassenaar, 2018*). Typically, migratory connectivity of neotropical migrants using stable isotopes can be estimated by capturing juvenile birds during migration or on the non-breeding grounds and using probabilistic statistical techniques to estimate their geographic locations on the breeding grounds where they grew their feathers. This works well because North American continental hydrogen isoscapes generally have a large latitudinal gradient and are, in general, better defined than those of tropical locations typical of non-breeding sites.

For this study, however, we tested an alternate approach to see if any evidence of migratory connectivity could be observed by measuring hydrogen isotopic compositions of adult birds at different locations on the breeding grounds to infer their wintering locations. Ruby-throated Hummingbirds, unlike many other Nearctic-Neotropical migrant passerines, molt flight feathers including rectrices on the non-breeding grounds (*Baltosser, 1995*), Therefore, a tailfeather collected from an adult hummingbird on the breeding grounds will reflect the isotopic signature of the non-breeding area, allowing an estimation of wintering range by probabilistic assignment to origin (*i.e.* isoscape mapping). The impetus for this approach was that bird capture and access is considerably less expensive and logistically simpler in Canada and the USA than in the remote areas of Mesoamerica. Additionally, sampling is more efficient because adults generally form a higher proportion of captured birds.
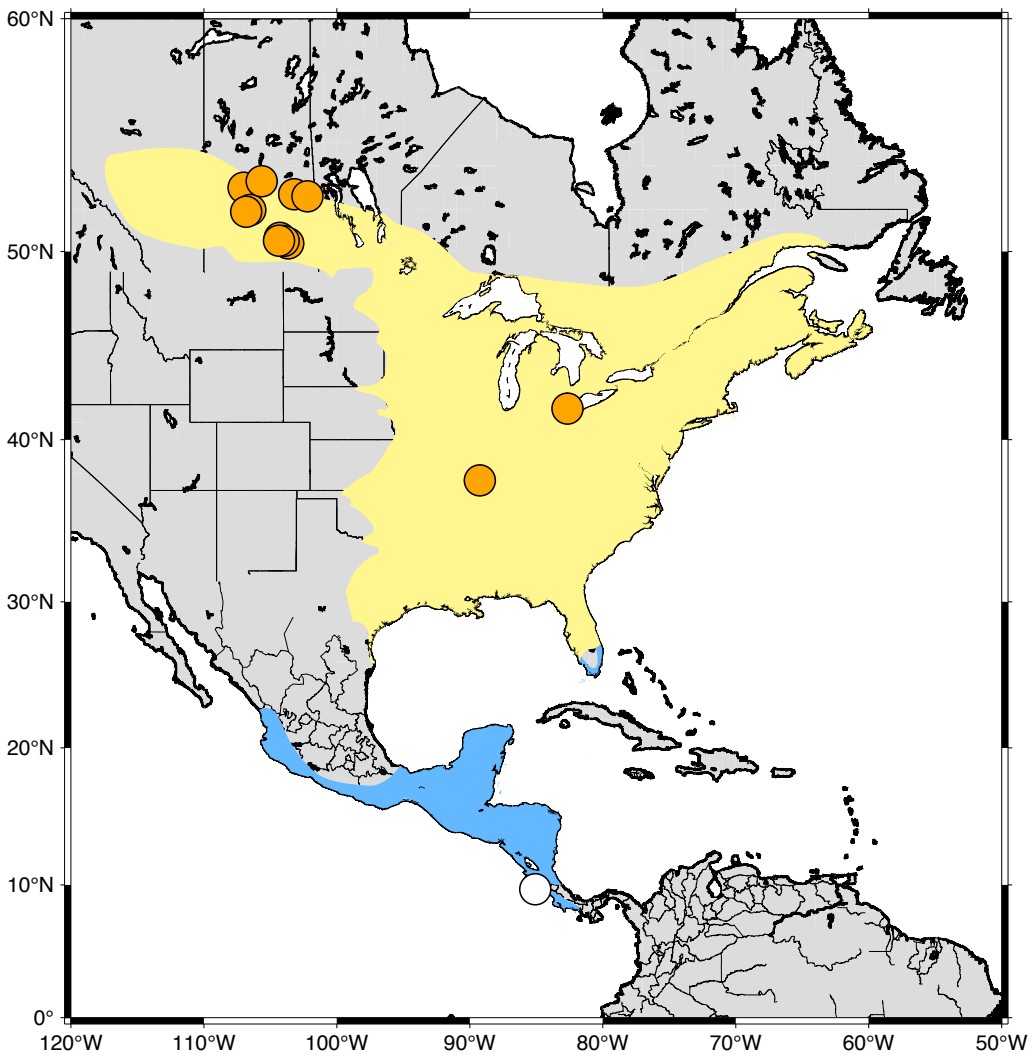

**Figure 1** **Sampling locations of Ruby-throated Hummingbirds in this study.** Yellow area indicates breeding range while blue indicates wintering grounds. Sampling sites are indicated by orange (North America) and white (Costa Rica) circles.

Finally, we measured both the hydrogen and oxygen stable isotopic compositions of hummingbird feathers to evaluate the relationship between $\delta^2H_f$ and $\delta^{18}O_f$ in this species, and to determine if their combined use might increase the accuracy of provenance estimates. Such measurements are rare in the literature (*e.g., Wommack et al., 2020*; *Hobson & Koehler, 2015*; *Koehler & Hobson, 2019*; *Hobson, Kardynal & Koehler, 2019*) and consequently the mechanisms responsible for routing of oxygen isotopes from environmental water to animal tissues remain largely unknown.

## EXPERIMENTAL

### Sample collection

Feather samples were collected from male and female adult hummingbirds during 2019 bird banding operations from June-Aug in Saskatchewan (SK), June in Ontario (ON), and July in Illinois (IL) (Fig. 1). Hummingbirds were captured using a hummingbird feeder placed inside a modified Hall Trap (*Jensen, 2017*) or a Dawkins passive trap. Captured hummingbirds were immediately removed from the trap, banded, and a single R4 rectrix feather was pulled from each bird prior to release. We sampled the R4 rectrix to minimize the effect of sampling on bird flight and to maintain consistency with previous studies (*Wassenaar, Hutcheson & Hendrix, 2007*; *Hutcheson, Hendrix & Moran, 2010*). Additional recorded parameters were band number or existing band number if banded, weight, sex, age, wing chord and culmen lengths, body fat, cloacal protuberance if female, and parasite type and number. In addition, we captured two juvenile birds in Cóbano, Costa Rica, at the southern range of their wintering grounds in January, 2020. This study was approved by the University of Saskatchewan Animal Research Ethics Board (EREB), certificate number 20190037. A summary table of collection details and isotopic results is presented in Table S1. Feathers were collected under Environment and Climate Change Canada banding permit 10815 AH (SK, Canada), banding permits 23266 (IL, USA), 10800 (SK, Canada), and 10755 (ON, Canada). Samples were imported into Saskatchewan under permit 131282 and 131044. Feathers were collected in Costa Rica with permit ACT-PIM-001-2020. All feather samples were shipped to the NHRC Stable Isotope Laboratory under permits ECCC 10815 AH and SK 131282.

### Stable isotope analysis

Hydrogen and oxygen stable isotopic compositions of all feather samples were measured at the NHRC Stable Isotope Laboratory of Environment and Climate Change Canada in Saskatoon, SK, Canada following the methods described in *Hobson & Koehler (2015)* and *Koehler et al. (2023)*. Briefly, feather samples were cleaned of adherent debris prior to analysis and any surface oils were removed by rinsing in 2:1 chloroform:methanol. Feather vane material was subsequently cut from individual feathers using clean stainless steel scissors, encapsulated in silver cups, and analysed using a Delta V Plus IRMS system (Thermo Finnigan, Bremen, Germany) equipped with a TC/EA device and a Costech Zero-Blank autosampler. We used Environment Canada keratin reference standards CBS (Caribou hoof) and KHS (Kudu horn) to calibrate sample $\delta^2$H ($-197$ and $-54.1$ ‰, respectively) and $\delta^{18}$O values ($+2.50$ and $+21.46$ ‰, respectively (*Qi, Coplen & Wassenaar, 2011*)). This normalization with calibrated keratins also corrects for any hydrogen isotope measurement artefact caused by production of HCN (*Gehre et al., 2015*) in the glassy carbon reactor as described by *Soto et al. (2017)*. Based on replicate ($n = 10$) within-run measurements of keratin standards and from historical analyses of an in-house QA/QC reference (SPK keratin), sample measurement error was estimated at $\pm 2$ ‰ for $\delta^2$H and $\pm 0.4$ ‰ for values $\delta^{18}$O. All H results are reported for nonexchangeable H and for both H and O in the standard delta notation, normalized on the Vienna Standard Mean Ocean Water –Standard Light Antarctic Precipitation (VSMOW-SLAP) scale. We used legacy

values for the non-exchangeable hydrogen isotopic compositions of CBS and KHS to retain compatibility with previous analyses of RTHU feathers analysed in our laboratory in 2007 and 2010 (*Wassenaar, Hutcheson & Hendrix, 2007*; *Hutcheson, Hendrix & Moran, 2010*).

## Rescaling function

We combined the hydrogen isotope analyses of known location hatch year (HY) RTHU reported by two previous studies (*Wassenaar, Hutcheson & Hendrix, 2007*; *Hutcheson, Hendrix & Moran, 2010*) to examine the rescaling function between the hydrogen isotopic composition of RTHU feathers and those of precipitation. Isotopic analyses in these two early studies were done in our laboratory and regressed the $\delta^2H_f$ values of HY birds to latitude or mean annual $\delta^2H$ values of precipitation. For this study we considered it more appropriate to use updated growing season (GS) precipitation values because growing season hydrogen isoscapes were just being developed at the time of these two studies and have been continuously improved since. As such, they provide a more accurate and relevant representation of the stable isotopic compositions of precipitation that are incorporated into tissues, especially for those species that have seasonal life cycles (*Bowen & West, 2019*). Migratory birds, including hummingbirds, often depend on dietary components derived from shallow rooted terrestrial plants and therefore are, isotopically speaking, more likely to reflect growing season precipitation than mean annual precipitation or groundwater.

Feathers from two birds from Saskatchewan collected for this study had $\delta^2H_f$ values that were too low to be grown on the wintering grounds in Mexico or Central America. We assumed that these were misidentified HY birds, replaced their feathers on the breeding grounds, or were returning adult birds that retained some of feathers grown on the breeding grounds from the previous year. These two birds were both caught in August, 2019, making misidentification of a juvenile bird the more likely scenario. In any case, the $\delta^2H$ values of these two individuals would reflect growing season precipitation in Saskatchewan and were included in the calculations to recast the transfer function between the $\delta^2H$ values of RTHU feathers to growing season precipitation (Fig. 2).

Similar to mean annual values and latitude, we measure an exceptionally strong ($r^2 = 0.95$) relationship between $\delta^2H$ values of RTHU feathers and growing season precipitation (Fig. 2), more so than for other bird species (see *Hobson et al., 2012*). Similar results were seen by *Moran et al. (2013)* whereby $\delta^2H$ values of tail feathers of Rufous Hummingbirds (RUHU, *Selasphorus rufus*) corresponded well to those of mean annual precipitation. To better compare the hydrogen isotope precipitation–feather relationship between these two species, we regressed the measured $\delta^2H$ values of RUHU as reported in *Moran et al. (2013)* to those of growing season precipitation rather than the mean annual values (Fig. 2). This resulted in a transfer function with a weaker relationship ($r^2 = 0.76$), but was similar to our recalculated transfer function. For hummingbirds, this close hydrogen isotopic coupling between feathers and growing season precipitation is likely a result of their reliance on nectar, resulting in a simpler connection to the hydrologic environment (*Hobson et al., 2020*). Hummingbirds will also variably consume insects which can constitute a considerable portion of their diet, especially when flowering

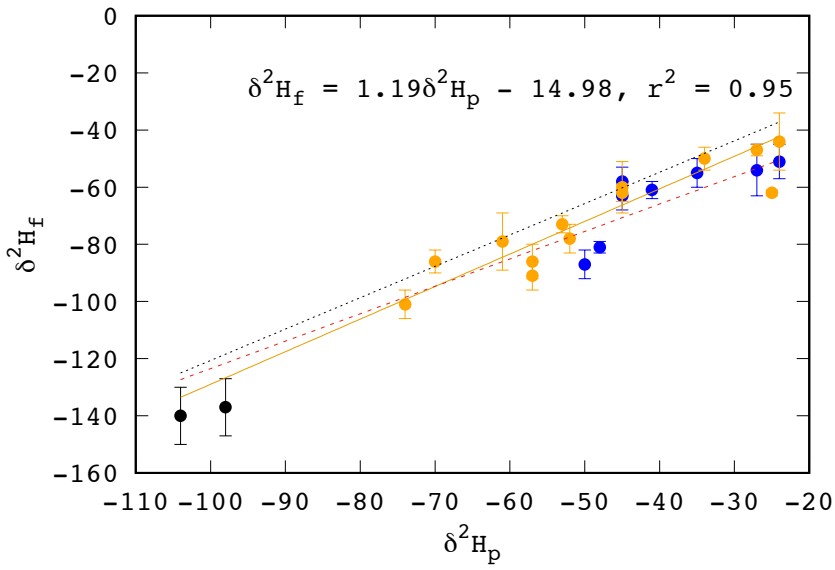

**Figure 2  Relationship between the $\delta^2$H values of feathers ($\delta^2$H$_f$) and growing season precipitation ($\delta^2$H$_p$) in HY Ruby-throated Hummingbirds.** Feather data and locations from *Wassenaar, Hutcheson & Hendrix (2007)* blue, *Hutcheson, Hendrix & Moran (2010)* yellow, and this study (black). Growing season precipitation $\delta^2$H values ($\delta^2$H$_p$) were extracted from the updated growing season isoscapes of *Terzer et al. (2013)*. Dotted line represents the recast relationship for Rufous Hummingbirds measured by *Moran et al. (2013)* between $\delta^2$H$_f$ and $\delta^2$H$_p$. Red dashed line is the relationship for aerial insectivores as determined by *Hobson & Koehler (2015)*.

plants are not available (*Montgomerie & Redsell, 1980*). However, local insects will also have hydrogen isotopic compositions that vary with the underlying hydrogen isoscape, so that it is difficult to separate the relative contributions of these two reservoirs to the hydrogen isotopic composition of RTHU feathers .

## Assignment algorithms

We used a likelihood-based assignment method as described by others (*Hobson et al., 2012*; *Wunder, 2012*) to depict potential non-breeding origins of individual hummingbirds. To accomplish this, we first created a species-specific $\delta^2$H$_f$ isoscape by re-calibrating known-origin RTHU samples from *Wassenaar, Hutcheson & Hendrix (2007)* and *Hutcheson, Hendrix & Moran (2010)* against the growing season $\delta^2$H$_p$ isoscape from *Terzer et al. (2013)*. We used the RTHU distribution map from BirdLife International (*Birdlife International, 2022*) representing the non-breeding grounds to delimit the spatial extent of the assignment areas. We applied a normalized probability density function (*Hobson et al., 2009*) to estimate the posterior likelihoods for individual cells in the calibrated isoscape representing a potential origin for each bird. We subsequently applied a conservative 2:1 odds ratio to the spatially explicit probability densities for our samples (*i.e.*, individual assignment rasters), where raster cells with ≥ 66.7% likelihood were coded as potential origins (1) and all other locations (*i.e.*, <66.7%) were considered as unlikely origins (0) (*Hobson et al., 2012*). This resulted in a binary raster file for each individual sample, which then were summed across assignments for all other individuals to represent potential origins

for each population. We chose this approach because individuals with high likelihood estimates can bias depictions of population-level (*i.e.* summed) assignments. Spatial analysis and assignment to origin analyses were conducted using the terra (*Hijmans, 2020*), maps, and assignR (*Ma et al., 2020*) packages along with routines re-written from the isocat package (*Campbell et al., 2020*) in the R statistical computing environment v.4.3.2 (*R Core Team, 2021*). We assumed a cutoff of $D$(Shoener's metric) = 0.20 to differentiate between similar origin surfaces (*Farmer, Cade & Torres-Dowdall, 2008*).

## RESULTS AND DISCUSSION

### Assignment to origin

Similar $\delta^2H_f$ values were measured for all birds from North America regardless of sampling location (Fig. 3). In fact, distributions of $\delta^2H_f$ values from birds collected from SK ($n = 64$), IL ($n = 24$), and ON ($n = 5$) were not statistically different in pair-wise comparisons (Welch $t$-test $p = 0.630$, $0.424$, and $0.648$ between SK-IL, SK-ON, and IL-ON, respectively). Feathers from the two HY birds collected from the non-breeding grounds in Costa Rica had $\delta^2H$ values of $-110$ and $-97$ ‰, consistent with those of previously measured HY birds collected from the breeding range. Contrary to the findings of *Moran et al. (2013)*, we observed no differences in $\delta^2H_f$ values between males and females captured in Illinois ($m = 5$, $f = 19$) or Saskatchewan ($m = 20$, $f = 44$) ($p = 0.887$).

Estimated non-breeding origins of RTHU for all sampled populations were similar but variable with the highest areas of potential provenance ranging from central Mexico, along the Pacific coast, and through the central part of its range in central Guatemala, southern Belize, northern Honduras and from most of Nicaragua. Consistent with the narrow range of $\delta^2H_f$ values, assignment of RTHU to their wintering range did not show any statistical difference between any of the assignment surfaces (Fig. 4), but we did detect a small difference between the assignment surfaces of birds from Saskatchewan and those of Ontario and Illinois. Schoener's D metric between SK-IL, IL-ON, and SK-ON assignment surfaces were 0.92, 1.0, and 0.92, respectively. In this instance, this result suggests that birds returning to Saskatchewan may have a slightly different distribution on the non-breeding grounds as do those from the more easterly populations in Illinois or Ontario. However, the sample size of the Ontario birds was small ($n = 5$) and the correlation between the SK and IL surfaces was still very good ($D = 0.92$) so that this result is tentative at best.

A major uncertainty involved in using isotopic methods to place birds on the non-breeding grounds is, in contrast to North American precipitation isoscapes, the relative homogeneity of the hydrogen isoscape in central America and Mexico. If most locations on the non-breeding grounds have similar hydrogen isotopic compositions of growing season precipitation and thus feathers, it would be difficult to tell the difference between birds from different locations that were captured on the breeding grounds. For example, with North American mean annual hydrogen isoscapes, it has been estimated that approximately 30 ‰ difference between sample distributions is needed for adequate separation (*Farmer, Cade & Torres-Dowdall, 2008*). Fortunately, there is a strong $\delta^2H_f - \delta^2H_p$ relationship for hummingbirds and standard errors of growing season isoscapes are generally lower

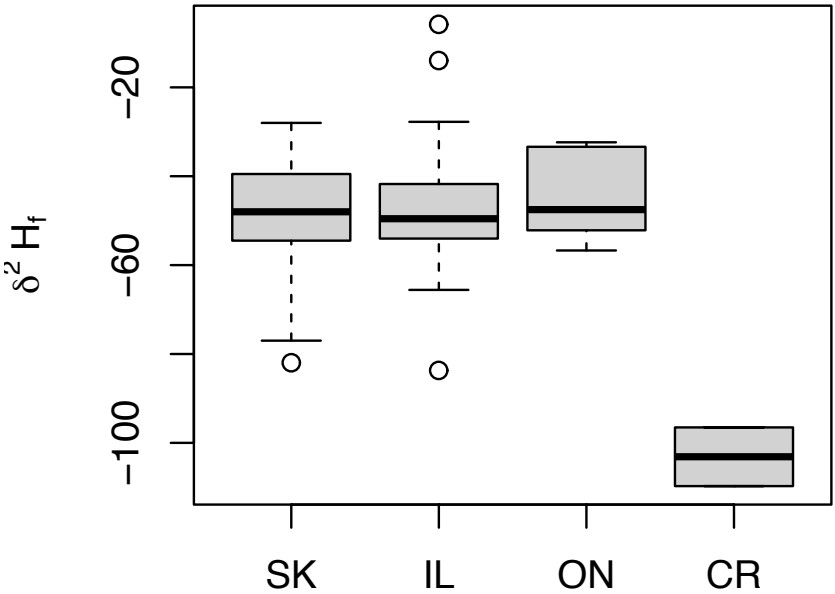

**Figure 3 Distribution of $\delta^2H_f$ values collected from adult Ruby-throated Hummingbirds collected from Saskatchewan (SK), Ontario (ON), and Illinois (IL) during the summer of 2019.** Also shown are $\delta^2H_f$ distributions of HY birds collected on the wintering grounds in Costa Rica (CR) January 2020.

than those of mean annual isoscapes. As a consequence, error estimates of the RTHU feather isoscape are close to the absolute measurement the error of about ± 2 ‰ (Fig. 5). Predicted $\delta^2H_f$ values differ by up to 30 ‰ on the winter feather isoscape and while this is the minimum separation for geographic placement, a strong migratory connectivity of our sample set, if present, should be detectable by the methods used. Other uncertainties include sampling bias (*e.g.*, IL birds were sampled on a single day), and we were only able to sample a single season because of pandemic restrictions in 2020.

Assignment surfaces of the breeding grounds from the two juvenile birds collected at the same location on the very southern part of the wintering grounds in Costa Rica were not significantly different ($r = 0.68, D = 0.65$) and placed these two birds in the northern USA and Ontario (Figs. 6A, 6B). This result is consistent with the inferred wintering areas from the adult birds collected at Pelee Island in Southern Ontario (Fig. 4C).

Similar to this study, *Hutcheson, Hendrix & Moran (2010)* collected juvenile RTHU from overwintering sites in El Salvador and Costa Rica. They reported differences in $\delta^2H_f$ values between the two sites consistent with chain migration. Interestingly, one of their collection sites was within 50 km of ours at Cóbano, Costa Rica and the $\delta^2H$ values were higher than our two samples ($\delta^2H_f = -57, -56$ ‰), inferring a different natal location than those of the birds we collected. Indeed, assigning these birds to their breeding location results in a more southerly placement (Fig. 6C) and incompatible assignment surfaces ($r = -0.6, D = 0.12$). The sample size is small but, in all, these observations augment those of the Ontario adult birds and suggest that there is a low degree of migratory connectivity

Peerl

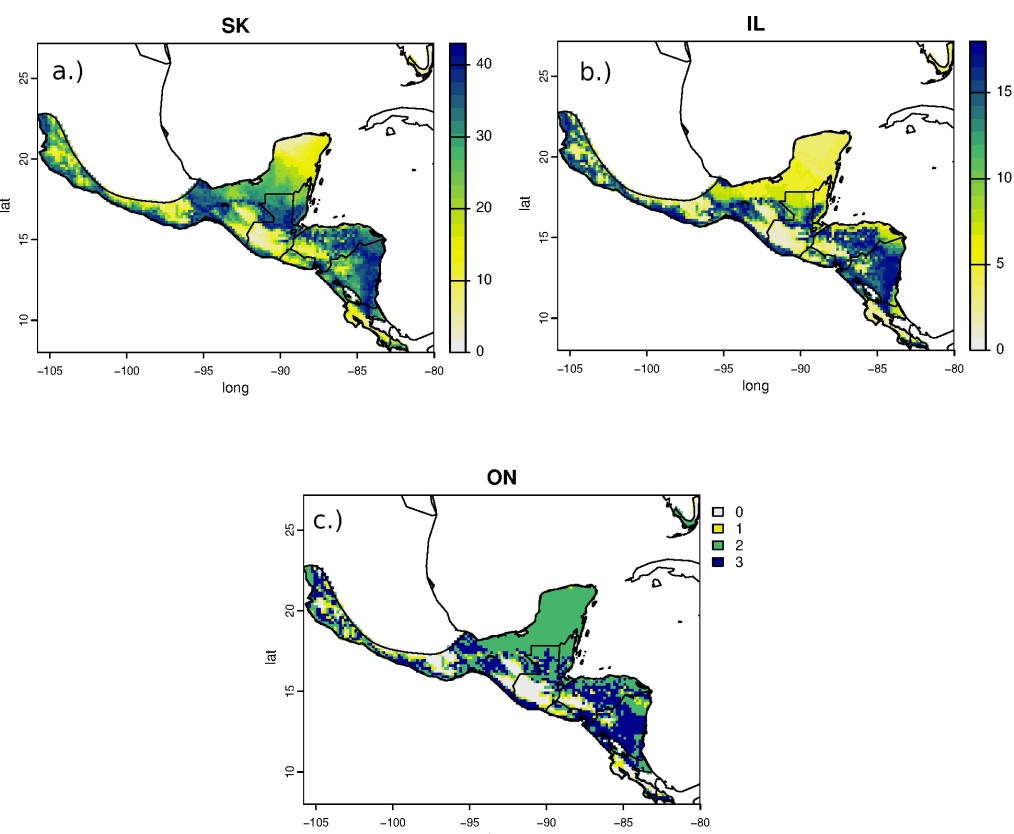

**Figure 4** Estimated non-breeding range assignments of adult Ruby-throated Hummingbirds collected from (A) Saskatchewan ($n = 64$), (B) Illinois ($n = 24$), and (C) Ontario ($n = 5$) based on feather $\delta^2H$ values. The legend indicates the number of birds that could potentially originate from each cell in the feather isoscape with a 2:1 odds cutoff.

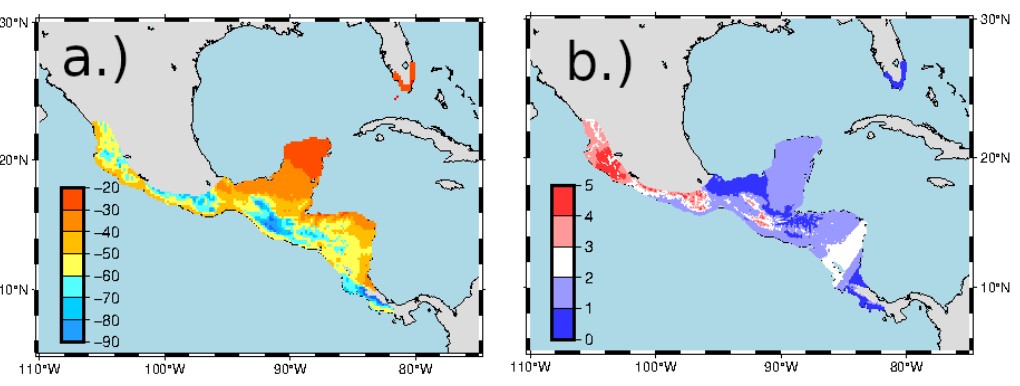

**Figure 5** Predicted $\delta^2H_f$ values (A) and standard errors (B) in ‰ for the non-breeding range of Ruby-throated Hummingbirds.

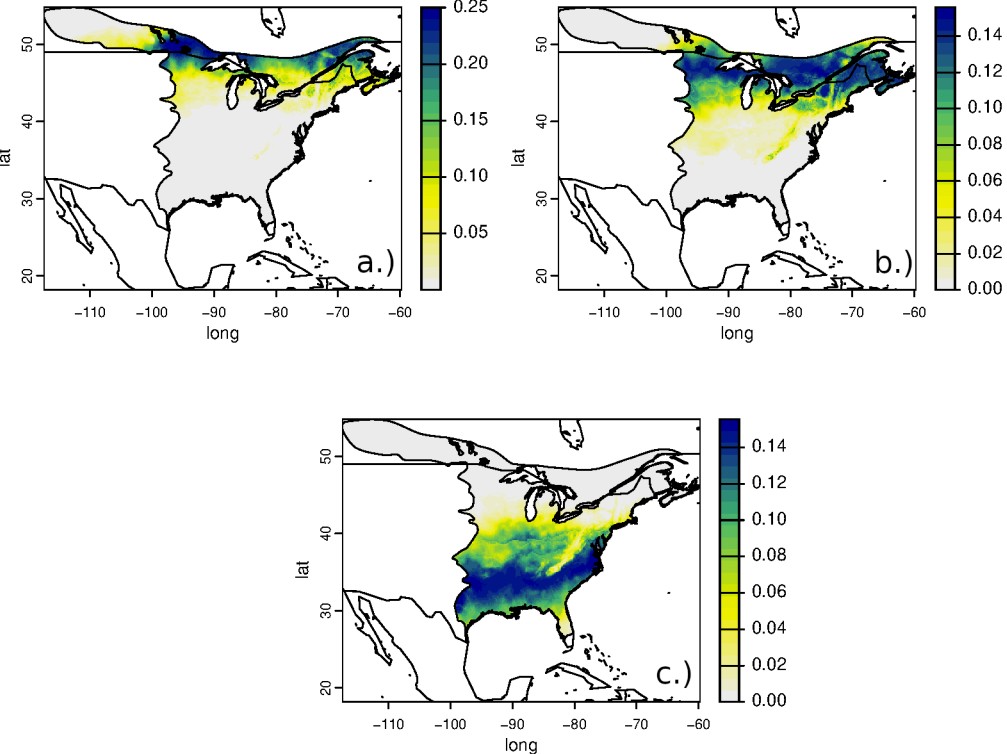

**Figure 6** **Predicted natal locations of juvenile RTHU collected at Cóbano, Costa Rica (A, B) and of those collected nearby by** *Hutcheson, Hendrix & Moran (2010)* **(C) based on $\delta^2$H values of feathers.** For clarity, cumulative probabilities are multiplied by 1,000.

between non-breeding and breeding grounds in this species, as also suggested by the inferred locations of the SK, IL, and ON adult birds.

Importantly, the outcome of 10 years of winter banding of three hummingbird species in the southern USA (*Bassett & Cubie, 2009*) suggest that hummingbirds have low wintering area fidelity, a conclusion that is consistent with our results from stable isotope methods. Despite our preliminary results, a robust determination of wintering and breeding fidelity for hummingbirds will require a concerted sampling and banding effort to determine recaptures and isotopic compositions of wintering hummingbirds at specific sites over multiple years.

## Oxygen isotopes

Oxygen stable isotopic composition of RTHU feathers ranged from +7.0 to +18.0 ‰. In stark contrast to hydrogen isotopes where the $\delta^2$H$_f$ values are highly correlated with those of precipitation, $\delta^{18}$O$_f$ values show poor correlation with their $\delta^2$H$_f$ values, ($r^2 = 0.14$) and therefore the $\delta^{18}$O values of precipitation (Fig. 7). Considering that $\delta^2$H$_f$ values were all within a small range, we would expect the same for $\delta^{18}$O$_f$ values, but clearly this was not observed. Similar results were seen for insectivorous passerine birds (*Hobson & Koehler, 2015*) and for monarch butterflies (*Danaus plexippus*) (*Hobson, Kardynal & Koehler, 2019*), but with a slightly higher correlation ($r^2 \approx 0.3$) between the $\delta^2$H and $\delta^{18}$O values. *Wolf*

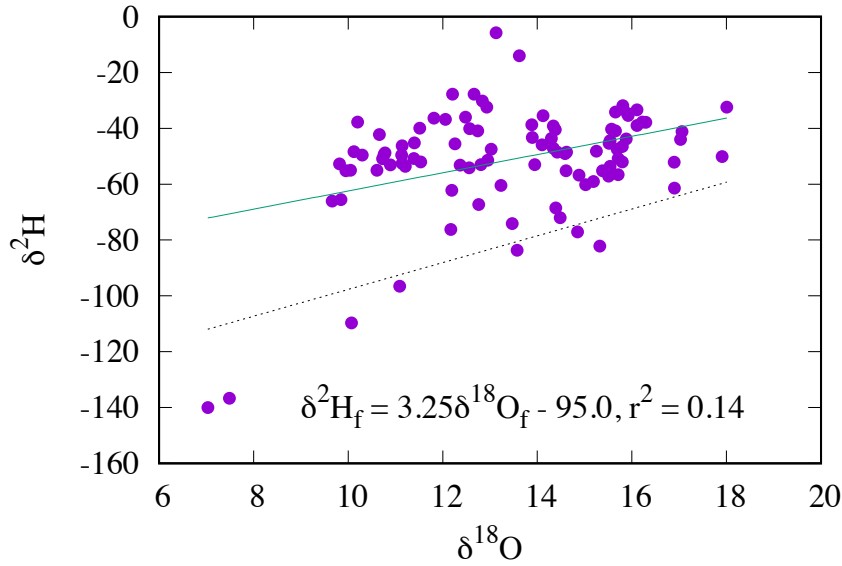

**Figure 7 Relationship between the $\delta^2$H and $\delta^{18}$O values of feathers in AHY Ruby-throated Humming-birds (solid line).** The dashed line is the $\delta^2$H–$\delta^{18}$O relationship for passerine insectivores (*Hobson & Koehler, 2015*).

*et al. (2013)* observed that while the $\delta^2$H values of feathers from captive Japanese quail (*Coturnix japonica*) reflected those of drinking water, no relationship was found for $\delta^{18}$O values. These results are in contrast to those of mammals, such as humans, cats, and dogs (*Ehleringer et al., 2008*; *Koehler & Hobson, 2019*), where the meteoric relationship between the hydrogen and oxygen isotopic compositions of hair are relatively robust.

Unlike hydrogen isotopes, the routing of oxygen isotopes from the environment to animal tissues is not only dependent on those of environmental waters and those of food, it is additionally complicated by respiratory processes where atmospheric oxygen is used for the processing of fats, proteins, and carbohydrates to produce energy (ATP) for metabolic processes (*Magozzi et al., 2019*). The catabolic products of respiration are $CO_2$ and $H_2O$ with the oxygen derived from both atmospheric oxygen and sugars or starches. For endotherms, such as birds and mammals, it has long been known that an inverse relationship exists between mass specific metabolic rate and body size so that smaller species must have a larger mass specific metabolic rate than large species in order to maintain constant body temperature (*Brody & Lardy, 1946*; *Schmidt-Nielsen, 1984*; *Taylor et al., 1981*). This means that for small bodied organisms the amount of body water produced by consumption of atmospheric oxygen during respiration must form a larger proportion of total body water than those of large bodied organisms. Consequently, isotopic models predict that because environmental waters form a larger proportion of the oxygen pool in large taxa, the $\delta^{18}$O values of body water should be more closely related to those of environmental waters (*Bryant & Froelich, 1995*). Conversely, for small bodied taxa, such as hummingbirds and passerines, this model may not apply because of the larger

relative effect of metabolic rates, respiration, and evaporative water vapour loss on the body water oxygen pool.

Hummingbirds have one of the smallest body weights of all avian taxa and an exceptionally high mass specific relative metabolic rate. Indeed, hummingbirds can reach mitochondrial oxygen consumption rates almost twice those of mammals at maximum $V_{O_2}$ (*Suarez et al., 1991*; *Suarez, 1992*; *Suarez, 1998*). Because it is known that high metabolic rates can additionally affect the fractionation of oxygen isotopes during respiration whereby the preferential incorporation of $^{16}O$ over $^{18}O$ is reduced at high metabolic rates (*Zanconato et al., 1992*), it is likely that the variably high metabolic rates of birds, especially hummingbirds, may further obscure or alter the isotopic composition of body water and thus feathers. Respiration requires diffusion of $O_2$ from alveolar cavities into the blood and mitochondria for the production of ATP. Diffusive processes are mass-dependent and thus oxygen isotope fractionation between lung membranes and blood will depend on the rate of $O_2$ consumption.

All things considered, it is evident that hummingbirds take the relative contribution of respiration and metabolism to the body water pool to the extreme with high mass specific $O_2$ consumption rates, lung $O_2$ diffusive capacities, and mitochondrial densities all coupled with low body weight. Even further complicating this, hummingbirds also display disparate metabolic rates, alternating between activity and topor where basal metabolic rates may correspondingly vary by an order of magnitude (*Powers, 1991*). Ultimately, this variable and large contribution of metabolically produced water serves to decouple the oxygen isotopic composition of the body water pool from the environment, although some coupling likely still exists, as the two feathers with the lowest $\delta^2H$ values also have the lowest $\delta^{18}O$ values (Fig. 7). Considering this weak relationship, it is impossible to calculate a transfer function between the $\delta^{18}O$ values of feathers and those of environmental water, necessary for geographic placement using oxygen isotopes.

## CONCLUSIONS

The hydrogen isotopic compositions of feathers and those of environmental water correlate exceptionally well for Ruby-throated Hummingbirds. The coefficient of determination ($r^2$) for this relationship was measured at 0.95 and is most likely a result the tight coupling between hummingbirds and nectar in the hydrological environment. Other than this, Ruby-throated Hummingbirds are observed to have a similar relationship between the hydrogen isotopic composition of feathers and environmental water as other birds.

Assignment using hydrogen isotopic methods of wintering grounds from North American adult Ruby-throated Hummingbirds as well as natal assignments of two hatch year birds from non-breeding grounds indicate that migratory connectivity is likely weak in this species for the populations that we sampled. Additional sampling in the eastern and southern parts of the species' breeding range as well as on the non-breeding grounds may provide greater insight into potential population-specific connectivity. However, the lack of a strong $\delta^2H_p$ gradient on the non-breeding grounds, coupled with sampling bias, and limited sampling seasons may also have resulted in the similarity of $\delta^2H_f$ values among sampled breeding populations.

Hummingbirds seem to be an extreme example of the relative importance of respiration to the oxygen isotopic composition of the body water pool, at least for small birds and other small endothermic vertebrates. Their unique physiology may serve to obscure or entirely decouple the oxygen isotopic compositions of environmental water with those of keratinous tissues. Consequently, although oxygen isotopes may be useful to geographically place mammalian species in certain circumstances (*e.g. Koehler et al., 2023*), these results indicate that oxygen isotopes may have limited utility to isotopically trace locations or migratory behaviours of birds in general and hummingbirds specifically.

## ACKNOWLEDGEMENTS

This study would not be possible without the expert help of master hummingbird banders: Cathie Hutcheson, Jared B. Clarke, and Cindy Cartwright. The efforts of Tyler Christensen, research coordinator, Wildbird Research Group Incorporated and Anthony Squitieri made collections in the Costa Rica banding site possible. We would also like to thank all of the bird banding hosts and office staff for their contributions. Sample preparation in the stable isotope laboratory was completed by K Courtney and J Fehr.

### Funding

This work was funded by Environment and Climate Change Canada. There was no additional external funding received for this study. The funders had no role in study design, data collection and analysis, decision to publish, or preparation of the manuscript.

### Grant Disclosures

The following grant information was disclosed by the authors:
Environment and Climate Change Canada.

### Competing Interests

The authors declare that they have no competing interests.

### Author Contributions

- Geoff Koehler performed the experiments, analyzed the data, prepared figures and/or tables, authored or reviewed drafts of the article, wrote the initial manuscript, and approved the final draft.
- Kevin J. Kardynal analyzed the data, authored or reviewed drafts of the article, and approved the final draft.
- Ron E. Jensen conceived and designed the experiments, performed the experiments, authored or reviewed drafts of the article, and approved the final draft.
- Keith A. Hobson analyzed the data, authored or reviewed drafts of the article, and approved the final draft.

## Animal Ethics

The following information was supplied relating to ethical approvals (i.e., approving body and any reference numbers):

The feather collection protocol was approved by the Research Ethics Board of the University of Saskatchewan #20190037 and collected under Environment and Climate Change permit 10815 AH.

## Field Study Permissions

The following information was supplied relating to field study approvals (i.e., approving body and any reference numbers):

Ontario samples were imported into Saskatchewan under permit 131282 and 131044. Feathers were collected in Costa Rica with permit ACT-PIM-001-2020. Feathers were collected in IL, USA under permit number 23266. All feather samples were shipped to the NHRC Stable Isotope Laboratory under permits ECCC 10815 AH and SK 131282.

## Data Availability

g-koehler. (2025). g-koehler/RTHUassignment: RTHU assignment code (RTHU_R1). Zenodo. https://doi.org/10.5281/zenodo.14911613.

## Supplemental Information

Supplemental information for this article can be found online at http://dx.doi.org/10.7717/peerj.19252#supplemental-information.

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
