# Peer review of "An evaluation of migration fidelity of Ruby-throated Hummingbirds inferred from stable isotope methods"

_PeerJ, doi:10.7717/peerj.19252_

## Round 0.1 · original submission · Major Revisions

The comments from the reviewers are detailed and should be useful in revising the manuscript. Both reviewers had concerns with lack of detail in the Methods, use of the d18O data, and code availability and statistical analysis. R2 also provides suggestions for how to improve the focus and rationale of the study and points out that details are missing regarding a permit for collection of some of the samples. In your revisions, please explain how you have addressed or why you have not addressed each reviewer comment.

Reviewer 1 ·

Basic reporting

General comments

My major comments are 1) Some of the methods, particularly those in the Assignment Algorithms section, are missing sufficient detail. 2) The 18O data are not used in assignments and therefor don’t seem relevant to the manuscript. I suggest removing these data from the manuscript. 3) I strongly suggest including R code in the Supplementary Material. This is now considered best practice for transparency in research. 4) There are numerous typos or missing words. Please give a careful proofread.

Specific Comments

Line 15: Although philopatry has been used in ecology to mean a return to a site of birth or hatching, its literal meaning is “liking or loving the fatherland”. In the age of sensitivity to terminology, I suggest changing this term to “site fidelity”.

Line 17: Is suggest inserting at beginning the sentence that begins on this line something like:” We measured stable hydrogen and oxygen ratios in tail feathers (R4)…”

Line 20: Please change “does” to do”.

Line 29: I suggest changing “other population” to “additional populations” since the implication is that sampling more than three breeding populations may reveal migratory connectivity.

Line 38: Please change “north American” to “North American.”

Line 40: Please delete “similarly” from this sentence and instead mention that these are breeding population declines. Are these range wide declines or are they reported form certain sites?

Line 42: Are habitat loss and deforestation different processes in this context?

Line 44: Not all migratory connectivity research aims to identify areas used during the non-breeding period. For species that undergo moult on breeding area, the reverse is true. For accuracy, I suggest limiting your definition of migratory connectivity to determining spatial linkages between breeding and non-breeding grounds.

Line 45: I suggest not using the descriptor “overwintering” because many Southern hemisphere migrants undertake their northbound migration in what they term “summer”. Please instead use “non-breeding” throughout. Also please change “are” to “is”.

Line 48” Please offset “including hummingbirds” with commas.

Line 52: Please inset a comma before “which”.

Line 59: I don’t think control is the correct word here. Perhaps “pattern”?

Line 63: One line 37, you list in parentheses “RTHU” after the common name. indicating that you will use the abbreviation going forward. I therefore suggest using “RTHU” here.

Line 64: Arctic-Neotropical should be capitalized and is more conventionally, “Nearctic-Neotropical”.

Line 65: “adult bird on the breeding grounds” for clarity.

Line 86: Change “Feather” to lowercase.

Line 104: Please define “HY”.

Fig2: This figure has a regression line for aerial insectivores. Is this relationship relevant to this study? It isn’t mentioned in the results.

Line 126: Please separate “tailfeathers”.

Lines 116-134: These lines report results and should therefore be better presented in the Results and Discussion rather than the Methods.

Lines 136-145: The assignment algorithms section needs more detail. I read the methods in Van Wilgenburg and Hobson (2011) and the approach in this paper differs substantially from this manuscript. For example, Van Wilgenburg and Hobson (2011) uses a normal probability density and Bayes rule to calculate likelihood of assignments of feathers with prior information about banding resights. They then convert likelihoods to probabilities using a 3:1 odds ratio. Since you have no prior information, and therefore do not use Bayes Rule, it does not make sense to cite this paper as grounds for not provide a more complete description of your methods. Some things that should be addressed: Did you rescale the del2F isoscape in AssignR or in another package? You mention custom R scripts. I strongly urge you to include your R code in the Supplemental Material for transparency (line 140). Specifically what spatial statistics were performed (line 141). The normal probability density will produce likelihoods, not probabilities if done according to Hobson et al. (2009). Be explicit about how probabilities were calculated form likelihoods (line 143). If you used an odds ratio to do this, which one? In what package were the resulting assignment probabilities visualized after this step? Finally, there is no information about how assignments were done with 18O.

Line 157: You cannot diagnose a” statistical difference” between two assignment surfaces just by visualizing them.

Line 159-160: Please present p-values for Pearson correlations.

Figure 5: This figure del2Hf map rescaled form del2Hp. If this map is to be used to argue that estimating connectivity is theoretically possible given the variation in del2Hf, then this figure should appear before Fig.4, the assignments.

Line178: You can’t do statistics on an n=2 sample.

Lines 178-180: It is tenuous to say that these two feathers from the non-breeding grounds don’t align with assignments from feathers sampled Ontario because only 5 feathers were sampled in Ontario, thus it is unlikely that the winter distribution in this population is sampled completely. It is more likely that the two feathers from Costa Rica give addition perspective on the possible winter distribution of the Ontario population.

Line189-191: It is not clear how the assignment data provides information on low site fidelity because the data were all collected in a single year, 2019. Regarding explanations for low non-breeding site fidelity, there is evidence that hummingbirds engage in altitudinal migration (e.g. Rueda‐Uribe et al. 2024, Ecography). If RTHU engage in such movements, understanding variation in moult altitude would be helpful for determining is measuring connectivity is possible.

Line195: From the Abstract and Introduction, I expected that assignments would also be done with 180, but this does not seem to be the case. If these isotope data are not used for assignments, I do not see a reason to include the 18O data in the manuscript.

Experimental design

See above.

Validity of the findings

See above.

Additional comments

See above.

Reviewer 2 ·

Basic reporting

This paper addresses a timely topic of identifying potential origins of ruby-throated hummingbirds using a multi-isoscape approach. As such, I appreciated the authors efforts on this topic, as identifying geographic origins (and assessing for potential migratory connectivity in hummingbird populations). This could provide a useful tool to help improve our understanding of factors influencing population declines for this (and other) hummingbird species. That being said, there are a number of items that remain to be addressed in this manuscript.

The main topics for this paper were reasonably outlined in the abstract (although it was not quite clear “why” the given isotopes were selected for their usefulness in identifying migratory connectivity.) Additionally, the first couple of paragraphs in the introduction provided a clear rationale for the use of isotopes to assess geographic origins. However, these objectives were lost in subsequent of paragraphs, and I had some difficulty following the methods (and especially the Results and Discussion).

Other specific comments are provided below in section 4.

Experimental design

The focus of this paper is relevant & meaningful (to assess potential migratory connectivity in RTHU). However, in its current form, it's not clearly defined, nor is it entirely clear how this study helps close the existing knowledge gap.

For example, after reading the introduction, the main objectives seemed to focus less on determining migratory connectivity, and more as a methods paper (to assess whether feathers collected on the breeding ground could sufficiently assign geographic origin of wintering grounds (rather than breeding grounds). Since RTHU are somewhat novel in that the majority of flight and tail feather molt occurs on wintering ground, highlighting the benefits (and challenges) of assessing geographic origins for wintering birds with less-distinct isoscapes (or providing suggestions of alternative single or multi-isoscape approaches that could work better) could be beneficial to the reader. However, the current title focuses on migration fidelity, which is only mentioned a couple of times in the main text, and not until the 4th page of the Results and Discussion section. The authors even state on LN 192-193, that “despite or preliminary results, a robust determination of wintering and breeding range fidelity for hummingbirds will require a concerted sampling and banding effort…”, suggesting that the data presented in this study is not sufficient to actually infer fidelity, and a more appropriate title should be used..

The Methods are lacking in sufficient details (including sampling locations, dates, or information on statistical tests (that are only mentioned in the Results). Additionally, the assignment algorithms paragraph mentions custom R scripts and spatial statistics, but no information (or SI files) appear to be included that provide details on these steps (or allow for analysis replication).

Additionally (as mentioned in comments provided in section 4), please provide information on relevant banding and collection permits (especially since this included samples from multiple countries). It appears all samples were analyzed in Canada, which would require shipping samples across multiple international borders). However, permit information was only supplied for the Canada samples (not the ones from Illinois).

Validity of the findings

The Results and Discussion was sometimes challenging to follow (and wandered a bit). There was no clear presentation of sample totals (other than LN 149-150) or results. Additionally, the authors repeatedly referenced “statistical significance” of the different populations, and presented results from multiple statistical tests (e.g. Welch t-test). However, none of these tests were mentioned in the methods section, including the data used for these comparisons, how the authors obtained these results (including whether they only used isotope values, or if you involved the posterior probability maps obtained from their isotope analyses), or how these results should be interpreted in the context of their paper (or hummingbird conservation). A flowchart highlighting the analysis steps (and the statistical tests used) would help clarify the process to the reader. For example, why did you use a Welch t-test to compare d2H from multiple populations (or which 2-populations were compared??)

There’s also no clear distinction as to “how” the authors compared assignments of different groups (other than just basic t-tests comparing raw d2H values). R packages like “AssignR” (or more recently “isocat”) allow for post-hoc analyses to assess posterior probability distributions, but it’s not clear what methods the authors used to assign their distributions (other than just estimating the posterior probability map for each individual) or to compare those resulting values across state/provinces.

Regarding data, Table S1 (not cited in the main document) did include the raw isotope values for the study. However, there was no clear information regarding each location (no obvious metadata to identify each location, or the associated state/province), and the total samples in the raw dataset do not appear to match the totals provided in the manuscript, but it's not clear where the discrepancies are coming from.

Additional comments

The main topics for this paper were reasonably outlined in the abstract (although it was not quite clear “why” hydrogen and oxygen isotopes were selected for their usefulness in identifying migratory connectivity.) However, these objectives were lost in the text after the first couple of paragraphs, and I had some difficulty following the methods and subsequent results/discussion. For example, after reading the introduction, the main objectives seemed to focus less on determining migratory connectivity, and more as a methods paper (to assess whether feathers collected on the breeding ground could sufficiently assign geographic origin of wintering grounds (rather than breeding grounds). Since RTHU are somewhat novel in that the majority of flight and tail feather molt occurs on wintering ground, highlighting the benefits (and challenges) of assessing geographic origins for wintering birds with less-distinct isoscapes (or providing suggestions of alternative single or multi-isoscape approaches that could work better) could be beneficial to the reader. However, the current title focuses on migration fidelity, which is only mentioned a couple of times in the main text, and not until the 4th page of the Results and Discussion section. The authors even state on LN 192-193, that “despite or preliminary results, a robust determination of wintering and breeding range fidelity for hummingbirds will require a concerted sampling and banding effort…”, suggesting that the data presented in this study is not sufficient to actually infer fidelity, and a more appropriate title should be used..

Similarly, the 2nd paragraph (e.g. LN 53-54) establishes that both hydrogen and oxygen are useful for assigning geographic origins of birds, but later in the results and discussion, the authors state that oxygen isotopes are poor predictors of geographic origin for birds (and especially hummingbirds). If this was one of the primary take home messages of the paper (there are >2 pages on this topic in the results/discussion), it would be useful to establish that one of the major objectives was also to assess whether oxygen was a suitable isotope to use for geographic assignments in small birds (with RTHU as a case study for a species with a high metabolic rate).

As a side note, even though the authors imply that oxygen isotopes may work poorly for birds in general there are other studies in the literature that have used both H and O isotopes to assess geographic origin in birds (e.g. Wommack et al. 2020, https://doi.org/10.1371/journal.pone.0226318 for one). It would have been useful for the authors to cite these other studies, or potentially present their results in a similar manner—by presenting a probability map for oxygen (compared to their d2H maps), to better highlight “why” oxygen may not be an effective assignment tool (at least for this species).

Of larger issue, given the increasing use of multi-isoscape approaches to assess geographic origins of birds (and their wintering grounds—e.g. Hobson and Kardynal 2016 and Hobson et al. 2012 (https://doi.org/10.1890/ES12-00018.1), it’s not clear why the authors did not mention or consider any of these approaches (or use an alternative isotope (other than oxygen) that might be more suitable for these analyses). The lack of any reference to multi-isoscape analyses seems a bit short-sighted in this regard.

Another limitation of this study was the data itself. It was not clear in the main text as to when and where samples were collected (other than a vague “Summer 2019”) and Figure 1 (without referencing sample sizes, except for LN 149-150), or the geographic relevance of each of these locations to the overall study (i.e. are these known populations of interest? Do these locations provide a representative sample of the species --and your ability to adequately assess migratory connectivity)?

Table S1 does provide a list of all the samples, but it was also challenging to read, with no clear sort order, and it was almost impossible to figure out which samples came from which state/province (other than the Illinois samples, listed on multiple pages), and the 2 Costa Rica samples, only clearly identified based on their d2H values. The results section could benefit from reordering, and providing a clear summary table in the main document including the sample size totals, province/state of each survey location, and summary stats, etc. I spent a lot of unnecessary time searching through the text in the results to find this information, which detracted from the paper. Additionally, a single season of data seems a insufficient to be able to assess fidelity, especially if one does not have any information on return patterns in subsequent monitoring years.

Finally, although the Conclusions paragraphs provide a adequate summary of the data (and a more concise summary of the discussion text addressing the difficulties in assigning geographic origins for RTHU), I was left with limited clarity about future analyses from the authors, or (perhaps more importantly), no suggestions on what best practices “could” be implemented to estimate geographic origins for avian species that may have atypical molt patterns, high metabolism, and wintering ranges in areas with limited isotopic variation. Assessing geographic origins for species with declining populations is an essential component of improving our understanding regarding limiting factors for these species. As such, clarifying how your paper contributes to filling this knowledge gap would help strengthen the reader’s interpretation of the significance of your paper.
* * *
Other specific comments are listed below:
LN 15: Please clarify that this species is found in the eastern half of the continent
LN 17-18, the sentences make no clarification as to “why” you are sampling these birds (or what you are sampling.) Providing text to state what you were specially measuring multiple isotopes to assess migratory connectivity (e.g. d2H and d18O), rather than relying on LN 23 to first introduce the isotopes sampled.

LN 20 :Lack of statistical difference does not necessarily imply no biological difference. Please clarify this distinction in the text.

LN 39: Canada and USA is broad, please clarify that these are predominantly the eastern portions of both countries (and where non-breeding grounds are), as well as reference Figure 1, which provides a range map for this species.

LN 41: This sentence is confusing. Are you implying that these declines are from observable changes in breeding habitat? Changes in breeding success? It's not clear if you are only referencing population counts, or actual demographic studies, which would seem to be highly relevant in this case.

LN 48: “typically not useful for small birds”… It might be better to revise this to historically instead of typically, since tracking devices have been developed in recent years for small species (including hummingbirds.)

LN 61-64: It would be better merge this paragraph with the previous one, because the text (as currently written), is confusing. Additionally, you already state that isoscapes in northern latitudes are better than non-breeding, but then you rely on this inefficient isoscape for mapping.

LN 74: It’s not clear which IL samples were ones collected in 2019 vs those used from earlier studies. However, in either case, please provide relevant Federal/State/Province collection/banding permits as applicable (especially for Illinois). The permit information provided in the reviewer files appears to only include permits for Canada, not the US.

LN 74: “Summer” can be a broad term. Please specify actual survey months (including range of dates at each location, if these were different)-- was this as part of an existing study? Maps??
LN 77: were there any birds with multiple feathers collected (or multiple parts of the feather collected) to confirm that R4 is a reliable indicator of potential wintering location?

LN 80-81: As mentioned previously, please also provide information on relevant banding and collection permits (especially since this included samples from multiple countries (and it appears all samples were analyzed in Canada, which would require shipping samples across multiple international borders).

LN 101-102: were any of those previously sampled feathers available for repeat analysis to match results (or to sample for d18O?)

LN 110-112: Do you have an appropriate citation to support your argument that they are more accurate and relevant representation”?

LN 113: Do you mean nectar? Or insects? If so, please clarify instead of this vague text.

Figure 2: please clarify which samples are from which study. (Perhaps using color). Also, revise color palette to be more color-blind friendly.

LN 116-121: This text belongs in results, not methods.

LN 119: How could they retain feathers grown on the breeding grounds if you already said they grew their feathers on the non-breeding grounds?? Or perhaps molt origins are inconclusive? Also, when were these birds captured? If it was early enough in the season, it should be obvious whether they were actually HY birds or not. (Note: I did observe that capture dates were listed in Table S1, but it would be useful to have this information in the main text (or to include a citation to table S1 here so the reader knows where to find this data).

LN 132: This diet statement is not reflective of what is suggested in the BOW account. Please clarify differences between the Hobson et al paper and the actual species account. (which suggests insects may account for up to 50-60% of diet at some times.) Which one is more reliable?

LN 137-138: “for predicting origins Ruby Throated Hummingbirds” Awkward sentence- please revise.

LN 140: is this R script provided as supplementary material so the analyses are reproducible? Also, please provide appropriate citations for all R packages, not just AssignR.
LN 142: This citation is in an odd format-- is this correct? (or is there an appropriate web location? (or DOI)?
LN 144-145: How did you determine differences in ranges? assignR does allow for Odd Ratio calculations to compare the posterior probabilities, but it’s not clear whether you used any comparisons other than t Tests on raw d2H values (and none of the statistical tests are listed in the methods).

LN 150: These sample totals don’t appear to match the totals provided in the Supplemental spreadsheet and table S1?
LN 150: As mentioned previously, there’s no prior mention of using these analyses to assess statistical difference in the methods (although they appear to be a major part of your paper, since the abstract specifically mentions lack of statistical differences.) Please add this information to the methods.
Also, please clarify if presenting these results which t-test values correspond to each pair—were you testing if each individual population differed from the others combined? Or comparing each population individually with the others (e.g. IL vs ON, etc). Based on the Figure 3 caption, it appears this “might” be what you were intending, but the text is not clear.

Figure 3—Figure axis text appears to be cut off. Additionally, please state in the figure caption that you excluded the 2 samples collected in SK with more negative d2H values (previously referenced in LN 116). Also, please include sample sizes on figures.

Figure 3: When were the birds captured in Costa Rica? (Again, this is only listed in Table S1, and not available in the main text).

LN 153: Provide citation for these previously measured birds.

LN 155: It’s not clear regarding the sample size differences between the 2 sexes at either location.

LN 154-155: Given all these comparisons in the text, it would be really valuable to have a summary table that includes the totals by location/sex, and mean d2H values.
The Table S1 appears to include some of this data, but I can’t find a reference to it in the main text, and there is no information provided regarding the locations provided in the table (including Province/State of origin, except for the Illinois samples (I think?) which are also not paired together.

Table S1- also related to this document, it’s not mentioned in the text, but many of these samples from a single location were birds captured in a single day (e.g. the Makanda, IL) How might this influence your results? Especially if you are trying to infer migratory connectivity for an entire location based on birds sampled from a single day?
The main text need to at least point out this potential bias in your sampling methods and how it could influence your data (and interpretation of those results).

LN 159: again, similar to the t-test, no mention why you used correlations here? When assign R has some built-in post-hoc analysis

Figure 4- Again (as before) this color range needs to be modified to be more color-blind friendly. Additionallly, there’s is no clarification as to what the scale bar on the plots references (I “think” it’s probability, but it’s not clear. Additionally, the figure abbreviations are unclear in the legend text (please explain all abbreviations, including lat, long, and State/provinces, as well as include sample sizes.

LN 173: A difference of up to 30 per mill seems to be a sizable difference in values and could lead to some substantially different estimated geographic origins. How could this influence the interpretation of your results?

LN 177-178: “Significantly different” from what? What statistical test did you use to compare these values ? And why do you say the results were “significantly different” when you provide no previous reference as to what your ‘significant” threshold actually is (e.g. alpha = 0.05? (Additionally,, these results are confusing, as r is typically a correlation value (as listed on LN 159), and 0.06 seems to indicate very poor correlation? Also, since the text mentions assignment surfaces, one would assume you compared the surfaces for these with other individuals, but it’s not clear here
Did you ever generate any mean summary surfaces for birds from the different populations (you mention comparing assignment surfaces elsewhere (e.g. LN 159-160), but I don’t know what methods you actually used.)
LN 178: I’m confused as to why you reference Figure 4-- is this supposed to be Fig 6? I also have no spatial reference for where Pelee Island is (or how many samples you collected from this location). It’s also not clear why you seem to only pointing out that these results are different than your adult birds (collected at a different location).
LN 182-83: Please provide values from the other study here for reference.

LN 193: I agree. Even 2 years of data would have provided a more robust dataset for you to work with. I’m not sure you can draw much of a conclusion with only 1 season of data (especially for something like fidelity)
LN 196-201: Why do you only mention d18O results in here at the end of this section? Or why did you not consider a dual isoscape approach (as mentioned previously? Or, (at the very least) why not generate probability density maps for both isotopes and compare/present the results from both. It could be highly informative to see how well (or how poor) the d180 maps compared to your d2H results. (and would provide support for your argument that d18O is less informative, especially for species like RTHU).
LN 238: Please provide relevant citations.
LN 265: The other study you reference here (Hobson and Koehler 2015), focused specifically on insectivorous songbirds, whereas, (as you highlighted in the previous paragraph), hummingbirds feed predominantly on nectar. So I’m not sure how directly comparable this might be? Unless there’s also information on how oxygen isotopic compositions may vary based on diet (i.e. insects v nectar?)
Also, there do appear to be other papers that have used both H and O isotopes in birds (e.g. Wommack et al. 2020, https://doi.org/10.1371/journal.pone.0226318 ,) Please cite these (and other relevant research), and provide context regarding how these studies fit within the context of your results (where d18O may not be as effective as d2H for geographic assignment).

LN 271: “Nicoya”-- Where was this location? If Costa Rica, please specify.

LN 272-275: No mention of State or Federal permits for the IL samples (or import permits to ship samples to Canada)?

---

## Round 0.2 · Minor Revisions

Thanks for thoroughly addressing and incorporating the feedback from the reviewers into the manuscript. I ask that the authors use the per mil symbol throughout the manuscript rather than writing "per mil". After that, I'll be able to accept it.

---

## Round 0.3 · accepted · Accept

Thanks for making the requested revision and for cleaning up the other minor issues.